# Evolutionary consequences of nascent multicellular life cycles

**Jennifer T Pentz[1]\*, Kathryn MacGillivray[2,3], James G DuBose[2], Peter L Conlin[2], Emma Reinhardt[4], Eric Libby[5], William C Ratcliff[2]\***

[1]Los Alamos National Laboratory, Los Alamos, United States; [2]School of Biological Sciences, Georgia Institute of Technology, Atlanta, United States; [3]Interdisciplinary Graduate Program in Quantitative Biosciences, Georgia Institute of Technology, Atlanta, United States; [4]Department of Biology, University of North Carolina at Chapel Hill, Chapel Hill, United States; [5]IceLab, Umeå University, Umeå, Sweden

**Abstract** A key step in the evolutionary transition to multicellularity is the origin of multicellular groups as biological individuals capable of adaptation. Comparative work, supported by theory, suggests clonal development should facilitate this transition, although this hypothesis has never been tested in a single model system. We evolved 20 replicate populations of otherwise isogenic clonally reproducing 'snowflake' yeast (Δ*ace2*/Δ*ace2*) and aggregative 'floc' yeast (*GAL1*p*::FLO1 / GAL1*p*::FLO1*) with daily selection for rapid growth in liquid media, which favors faster cell division, followed by selection for rapid sedimentation, which favors larger multicellular groups. While both genotypes adapted to this regime, growing faster and having higher survival during the group-selection phase, there was a stark difference in evolutionary dynamics. Aggregative floc yeast obtained nearly all their increased fitness from faster growth, not improved group survival; indicating that selection acted primarily at the level of cells. In contrast, clonal snowflake yeast mainly bene-fited from higher group-dependent fitness, indicating a shift in the level of Darwinian individuality from cells to groups. Through genome sequencing and mathematical modeling, we show that the genetic bottlenecks in a clonal life cycle also drive much higher rates of genetic drift—a result with complex implications for this evolutionary transition. Our results highlight the central role that early multicellular life cycles play in the process of multicellular adaptation.

**\*For correspondence:**
jenn.pentz@gmail.com (JTP);
ratcliff@gatech.edu (WCR)

**Competing interest:** The authors declare that no competing interests exist.

## Editor's evaluation

This study provides fundamental insight into the evolution of multicellularity by experimentally demonstrating that yeast strains that form clonal groups evolve stronger group traits than ones that aggregate into non-clonal groups. Through modeling and analysis of the mutations accumulated during the experiment, this paper provides compelling evidence in support of clonal development favoring selection at the group rather than the cellular level.

## Introduction

Multicellularity has evolved more than 50 times across the tree of life (*Umen and Herron, 2021*; *Grosberg and Strathmann, 2007*). Multicellular organisms vary widely in their life cycles, ecological niches, and traits, and there are few features shared by all members of this diverse group (*O'Malley, 2022*). However, it is generally accepted that nascent multicellular lineages start out relatively simple (e.g. as undifferentiated groups of cells), gradually evolving increased organismal integration and functionality (*Grosberg and Strathmann, 2007*; *Brunet and King, 2017*; *Pfeiffer and Bonhoeffer, 2003*; *Knoll, 2011*; *Herron et al., 2022*). In some cases, multicellular groups that gain the capacity

for group reproduction undergo a shift in evolutionary mode: the origin of multicellular Darwinian individuality (*Michod, 2007*; *Godfrey-Smith, 2009*; *Buss, 2014*; *West et al., 2015*; *Maynard Smith and Szathmary, 1997*). This shift, from cell-level traits being the focal point of adaptation in the unicellular ancestor, to group-level traits becoming the focal point of adaptation in a nascent multi-cellular lineage, represents a crucial tipping point in evolutionary dynamics. In this paper, we examine the impact of early multicellular life cycles on the capacity for groups to serve as evolutionary units capable of adaptation.

There are two basic routes to forming a multicellular body. Nascent multicellular organisms employing either of these routes to group formation possess distinct multicellular life cycles. Individual cells can 'stay together' by forming permanent cell-cell bonds, forming clonal clusters that exhibit little within-group genetic variation. Alternatively, free-living single cells can 'come together', or aggregate, and often evolve to do so in response to some stimuli, such as starvation (*Bonner, 1998*; *Bonner, 2001*; *Crespi, 2001*; *Tarnita et al., 2013*; *Velicer and Vos, 2009*). Multicellularity has evolved multiple times via both staying together and coming together (*Grosberg and Strath-mann, 2007*; *Bonner, 1998*; *Du et al., 2015*), but 'complex multicellularity' (*Knoll, 2011*; e.g. plants, animals, fungi, red algae and brown algae) has only evolved in lineages that develop clonally (*Brunet and King, 2017*; *Knoll, 2011*; *Fisher et al., 2020*). Evolutionary theory explains this observation as a consequence of social evolution: clonal life cycles are a simple and powerful mechanism that 'de-Dar-winizes' cells (*Michod, 2007*; *Godfrey-Smith, 2009*; *Godfrey-Smith et al., 2013*) while Darwinizing multicellular groups. By limiting within-group genetic diversity, clonal development prevents intra-organism genetic conflict, as there is little standing within-group genetic variation for selection to act on *Buss, 2014*; *Maynard Smith and Szathmary, 1997*; *Clarke, 2014*. Any genetic variation that arises due to mutation gets partitioned among multicellular offspring, allowing selection to act on the group-level effects of de novo mutations (*Dahaj et al., 2021*). Clonal groups align the fitness interests of cells and groups, allowing cells to evolve altruistic social traits necessary for cellular differentiation (*Michod, 2007*; *Michod, 2006*). Finally, organisms that aggregate from a free-living state may expe-rience contrasting selection on the fitness of free-living single cells and the fitness of these cells in a multicellular group (*Márquez-Zacarías et al., 2021*).

While the numerous origins of multicellularity provide a uniquely rich set of natural experiments to examine correlations between developmental mode and organismal complexity (*Umen and Herron, 2021*; *Brunet and King, 2017*; *Bonner, 2004*; *Fisher et al., 2013*), there is no direct evidence that clonal development has played a causal role in the transition to multicellular individuality, and the subsequent evolution of organismal complexity. Indeed, a number of alternative explanations exist. For example, clonal development has evolved more frequently in aquatic environments, while aggre-gation has evolved more often in terrestrial environments (*Bonner, 1998*; *Fisher et al., 2020*). The lower complexity of aggregative organisms may be due simply to different life history selection in these fundamentally different environments, rather than evolutionary constraints due to within-organism genetic diversity. Alternatively, the potential for cellular differentiation appears to be highly contingent on the cell biology and behavioral repertoire of the unicellular ancestor (*Brunet and King, 2017*; *Hanschen et al., 2016*; *King, 2004*; *King et al., 2008*; *Suga et al., 2013*). The fact that complex multicellularity evolved in just five lineages may reflect historical contingency in developmental plas-ticity in these specific lineages, rather than evolutionary consequences of early developmental mode. Progress has been limited by the fact that all known transitions to multicellularity occurred in the deep past (>200 MYA *Grosberg and Strathmann, 2007*; *Herron et al., 2009*), obscuring the evolutionary dynamics of early multicellular evolution in extant lineages. Experiments directly comparing the evolu-tionary consequences of developmental mode have not yet been conducted, in part due to a lack of experimentally tractable model systems that can be induced to undergo either clonal or aggregative development.

Here, we circumvent these constraints by synthetically generating an isogenic yeast model system capable of either clonal or aggregative development. We created clonal 'snowflake yeast' by knocking out *ACE2* in a unicellular ancestor (*Ratcliff et al., 2017*; *Pentz et al., 2020*), and aggregative floc yeast by placing the dominant *FLO1* gene under transcriptional control of the *GAL1* promoter (*Pentz et al., 2020*; *Smukalla et al., 2008*). While these strains differ only in these two genes, these differences produce two fundamentally different life cycles: obligately multicellular snowflake yeast undergo a unicellular genetic bottleneck during ontogeny, making them clonal (*Ratcliff et al., 2017*; *Ratcliff*

*et al., 2012*; *Ratcliff et al., 2015*), while floc yeast can form genetically diverse aggregates (*Smukalla et al., 2008*). We evolved 20 populations of each genotype for 24 weeks with galactose as the main carbon source, ensuring robust flocculation, selecting daily for both faster growth and increased multicellular size by selecting for rapid sedimentation in liquid media (*Ratcliff et al., 2012*). We chose this selective regime because it is a simple and powerful way to examine the evolutionary consequences of selection acting simultaneously on both cell-level traits (such as growth rate) and group-level traits (such as settling rate). While buoyancy regulation may be an important driver of multicellularity in some lineages (*Dudin et al., 2021*), we chose this selective regime because it is an efficient way to select on multicellular size, a fundamentally important multicellular trait (*Tong et al., 2022*).

In our experiment, both floc and snowflake yeast adapted to this fluctuating environment, settling faster and growing faster. However, competition experiments with their ancestors reveal fundamentally different modes of adaptation. In clonal snowflake yeast, groups of cells served as the primary evolutionary units, gaining nearly all their increased fitness during the settling selection phase of the experiment. In contrast, aggregative floc gained most of their fitness as the cell-level, with evolved floc growing faster but possessing no measurable advantage during settling selection when competing against their ancestor. Snowflake yeast, but not floc, are evolving as primarily multicellular Darwinian individuals. Clonal development, however, resulted in far more genetic drift than aggregation. Mathematical modeling suggests that this is the result of a key difference in their life cycles. In floc, a rare beneficial mutant can disperse into many groups, while in clonal snowflake yeast it is constrained to a few, exposing them to greater sampling error during group selection. This is a general property of clonal multicellular life cycles, and this disparity scales with the number of cells within the organism prior to reproduction. Together, our results show how a simple difference in the mechanism of group formation, whether cells adhere with reformable bonds or adhere permanently, can fundamentally change their subsequent evolutionary dynamics and impact the level of Darwinian individuality (*Rose and Hammerschmidt, 2021*).

## Results

### Experimental evolution

Our selection regime involved 24 hr of batch culture followed by daily selection for rapid sedimentation in liquid media (*Ratcliff et al., 2012*). This selective scheme has previously been shown to promote multicellular adaption in snowflake yeast (*Ratcliff et al., 2012*; *Ratcliff et al., 2015*; *Ratcliff et al., 2013*; *Bozdag et al., 2023*) and has led to increases in cluster size of up to 20,000-fold over 600 consecutive rounds of selection (*Bozdag et al., 2023*). We have previously quantified the effect of settling selection on snowflake yeast by using a variety of tools (i.e. microscopy and flow cytometry *Ratcliff et al., 2012*; *Ratcliff et al., 2015*; *Ratcliff et al., 2013*; *Bozdag et al., 2023*) that cannot be

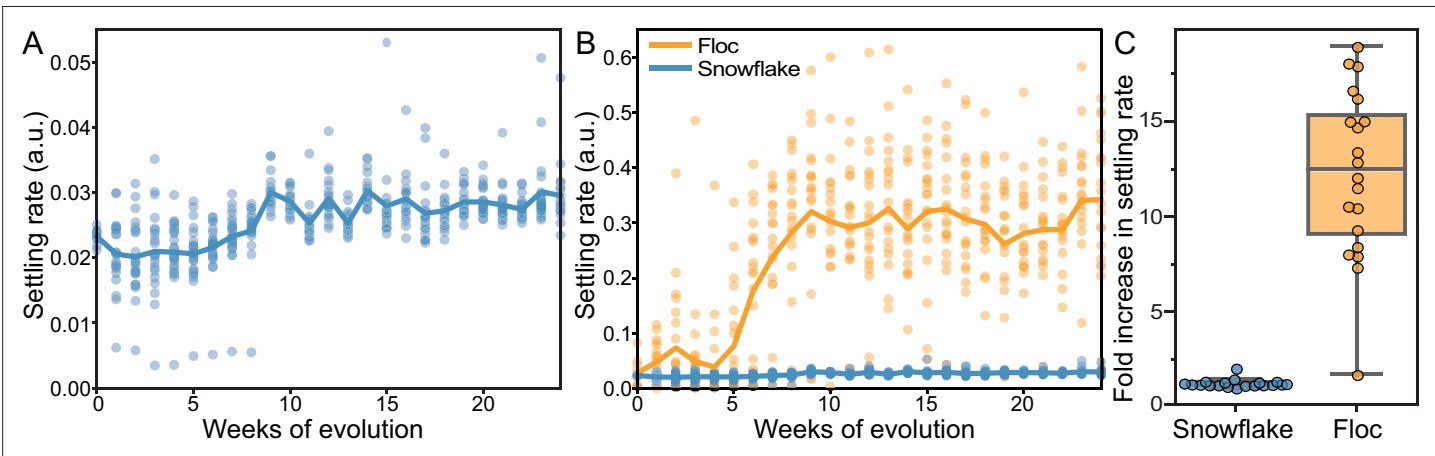

**Figure 1.** Dynamics of settling rate evolution in snowflake and floc populations. The settling rate (as determined by time-lapse imaging, see Methods) of 40 independently evolving populations of snowflake yeast (**A**) and floc yeast (**B**) over 24 weeks of evolution. Settling rate was measured for each population weekly. Each data point shows the mean of 4 biological replicates for each population. Snowflake yeast evolved to settle 30% faster over 24 weeks of evolution, while floc evolved to settle an average of 12-fold faster (**C**).

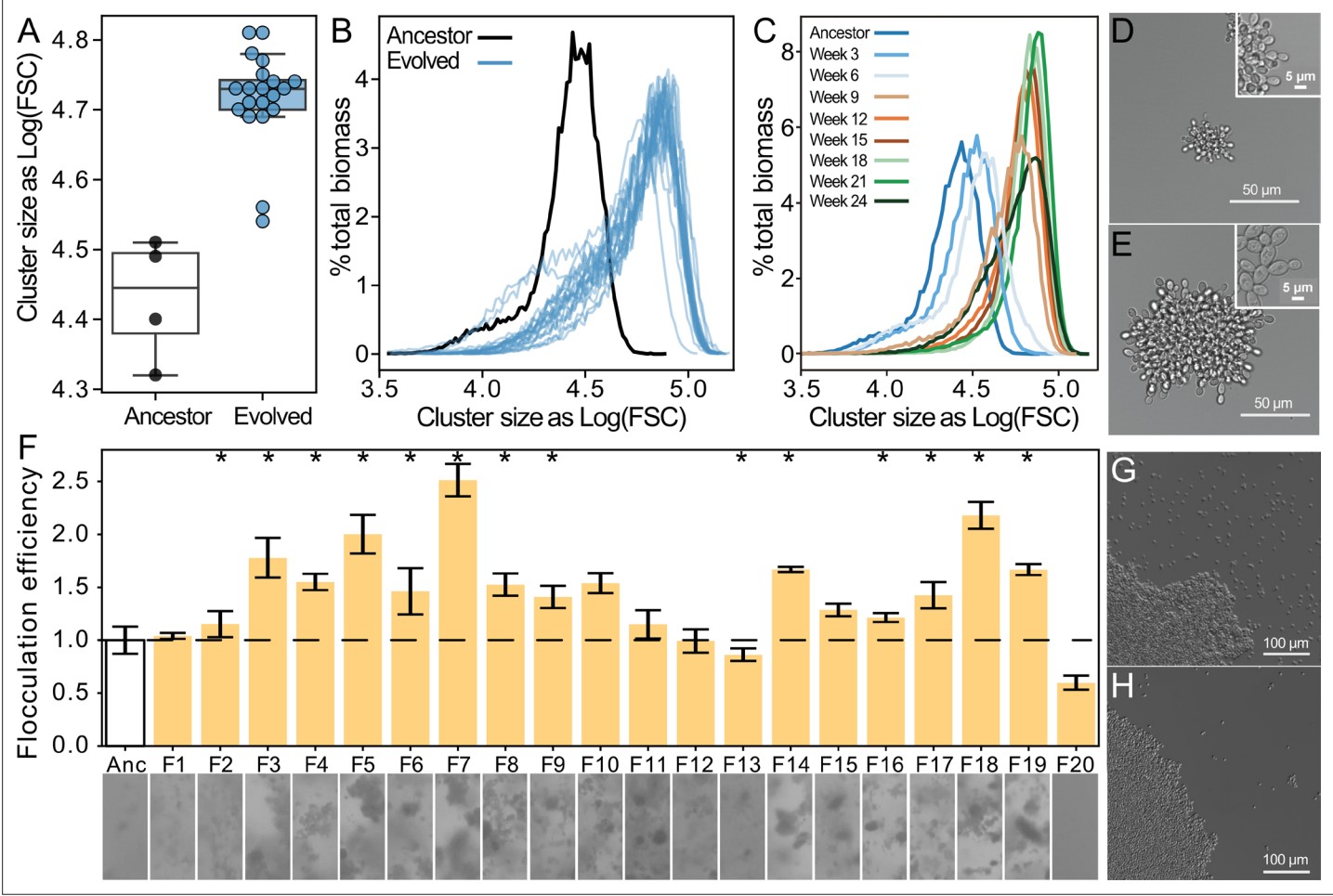

**Figure 2.** Both floc and snowflake yeast evolved to form larger groups. (**A, B**) Snowflake yeast increased their settling rate by evolving larger cluster size. Shown in (**C**) is the size distribution of a representative snowflake population (S8) every three weeks for the duration of the experiment. Relative to their ancestor (**D**), evolved snowflake yeast (**E**) are larger but retain a similar pattern of cellular attachment (insets). (**F**) We estimated the flocculation efficiency of ancestral and evolved floc by measuring the coefficient of variation in pixel intensity within cuvettes of a well-mixed population taken from shaking incubation (data on top, images from each population on the bottom). Flocculation efficiency was significantly higher in 13/20 evolved populations (significance at the overall $\alpha$=0.05 level denoted by asterisks). Shown are the mean and standard deviation of four biological replicates normalized to the mean of the ancestor. Representative images of ancestral (**G**) and evolved (**H**) floc yeast (genotype F4), showing fewer planktonic cells in the evolved isolate with higher flocculation efficiency.

The online version of this article includes the following figure supplement(s) for figure 2:

**Figure supplement 1.** Size distributions for evolved snowflake isolates.

**Figure supplement 2.** Flocculation efficiency is not dependent upon cell concentration of cultures.

**Figure supplement 3.** Clonal and aggregative yeast model system.

**Figure supplement 4.** Sensitivity analysis of biomass-transformation algorithm to bin sizes.

used for floc, because floc aggregates form dynamically as the clusters are settling. Thus, we developed a method to measure the settling rate of both floc and snowflake yeast populations during sedimentation (*Pentz et al., 2020*), calculating the displacement of biomass as they settle via high-resolution video (see Methods). We measured the settling speed of each of the 40 populations weekly over the course of the 24week experiment (*Figure 1A and B*). Both snowflake yeast and floc evolved to settle significantly faster (one-way ANOVA; floc: $F_{23,456}$=16.57, p<0.0001; snowflake: $F_{23,456}$=13.65, p<0.0001, pairwise differences assessed with Tukey's post-hoc HSD with $\alpha$=0.05). Floc, however, exhibited a much larger increase in settling rate than snowflake yeast (12-fold increase vs 1.3-fold increase after 24 weeks, respectively, *Figure 1C*).

We next examined the traits underlying increased settling speed. Using flow cytometry, we measured the size of 24-week evolved snowflake yeast. Biomass weighted mean forward scatter (a proxy for group size) increased by an average of 33% (*Figure 2A and B*; one-sided *t*-test, *t*=7.48, n=24, p<0.0001; non-overlayed histograms for individual snowflake isolates shown in *Figure 2— figure supplement 1*). Evolved snowflake isolates retain the same basic growth form (*Figure 2D and E*). Examining one lineage through time, we found that size appeared to plateau after 8–10 weeks of settling selection (*Figure 2C*), which is consistent with previous work in this model system where aerobic metabolism, and the corresponding reliance on diffused oxygen for growth, strongly inhibits the evolution of increased size (*Bozdag et al., 2023*; *Bozdag et al., 2021*). By 24 weeks, floc yeast appeared to be aggregating far more efficiently than their ancestors. To quantify this, we measured the coefficient of variation in pixel opacity in a well-mixed population just prior to settling selection (*Figure 2F*). This standardized variance measurement is as a proxy for flocculation efficiency, as when more cells are in flocs, the biomass will be more heterogeneously distributed within the cuvette. Thirteen populations showed significantly increased flocculation efficiency relative to the ancestor (one-way ANOVA, $F_{(20,46)}$=45.53, p<0.0001, pairwise differences assessed with Tukey's honestly significant difference [HSD] with α=0.05). This resulted in a noticeable reduction in the density of free un-flocculated unicells, see examples of the ancestor (G) and a representative 24-week isolate (H). We confirmed that increased flocculation was not a product of cell density in floc cultures (*Figure 2— figure supplement 2*) Thus, floc yeast evolved to settle more rapidly by forming larger aggregative groups and reducing the proportion of non-aggregated cells in the population (*Figure 2G and H*).

## Partitioning fitness between growth and multicellular-dependent survival

A common way to analyze ETIs is to use the Price equation to partition fitness arising from selection acting at distinct levels, i.e., cells and groups (*Rose and Hammerschmidt, 2021*; *Shelton and Michod, 2020*). Our system is not amenable to this kind of decomposition given the dynamic nature of flocs: groups rapidly form and fuse during settling selection, changing in size and composition until they either succeed at joining the pellet at the bottom of the tube, where they rapidly adhere to these cells, or fail to do so and are discarded. The fluid nature of flocs, and corresponding difficulty of measuring their traits without changing the traits we seek to measure, prevents us from quantifying the genetic composition and fitness of flocs during settling selection - data that is necessary for a Price

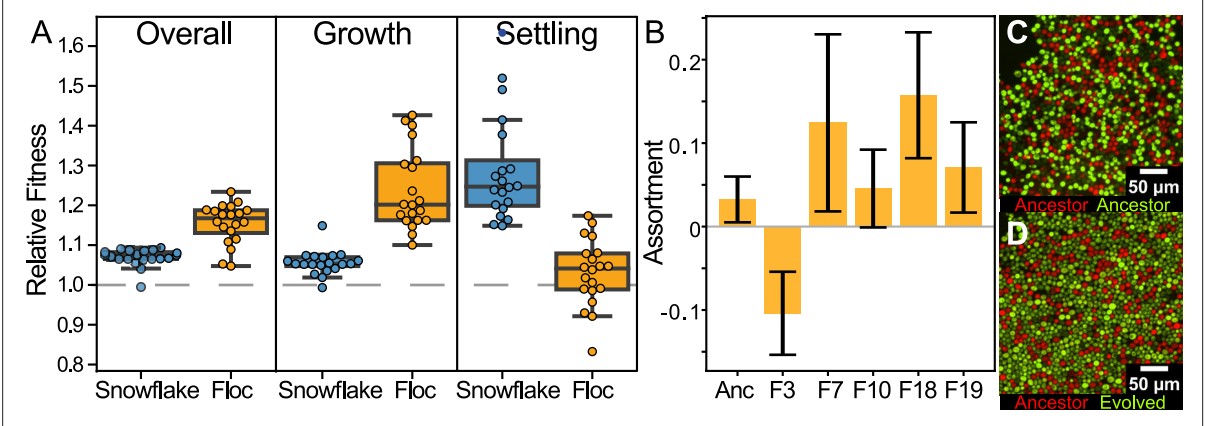

**Figure 3.** Examining fitness during growth and group-dependent competition. (**A**) When competing against their ancestors, both snowflake and floc yeast increased fitness in our experimental regime (first panel). Floc yeast had a 15% average fitness increase over three days with daily selection for settling, while snowflake yeast had a 7% increase. However, snowflake and floc yeast evolved distinct differences in their fitness during the growth and settling selection phases of our experiment. Floc yeast obtained a 30% fitness advantage during growth, but showed no detectable increase in fitness during settling selection. Alternatively, snowflake yeast had a 28% fitness advantage during settling but only a 6% advantage during growth. Data points show the mean of four biological replicates for each evolved isolate. (**B**) When competing against their ancestor, representative floc isolates formed groups with low genetic assortment. This may explain the decoupling between dramatically increased group sedimentation rate (*Figure 1B*), and the negligible increase in fitness during settling selection. Shown are the mean and standard deviation of three and six biological replicates for the evolved isolates and ancestor, respectively. Representative flocs are shown in (**C&D**), with the ancestor competing against itself, or a RFP-labeled ancestor competing against an GFP-labeled 24-week evolved isolate (genotype F10).

equation fitness decomposition. Fortunately, we can still gain insight into how selection is acting on cell and group-level traits by leveraging the biphasic nature of our experiment.

Fitness in our experiment depends on growth rate during the 24 hr of batch culture when the population is competing for resources, and survival during the subsequent settling selection step (*Pentz et al., 2020*; *Ratcliff et al., 2012*; *Conlin and Ratcliff, 2016*). Growth rate during batch culture is a classic trait under strong selection during experimental evolution (*Lenski et al., 1991*). The main way that a lineage can increase growth rates is by increasing the rate at which cells divide. In contrast, the main way in which a lineage can increase survival during settling selection is through changes in the phenotype of multicellular groups (i.e. increasing their size, packing density, or hydrodynamic profile *Ratcliff et al., 2013*). Thus, if populations evolve higher fitness during the growth phase, this can be taken as evidence that selection is acting effectively at the cell-level. Correspondingly, fitness improvement during the settling phase, where strong between-group selection determines survival, can be taken as evidence that selection is acting at the group-level.

We measured the fitness of evolved isolates (one isolate per 24-week evolved population, or 40 isolates in total) in competition against their unevolved ancestor one 24 hr period of growth, and one round of settling selection, allowing us to partition fitness increases amongst the two factors that affect fitness in our system: growth and settling. Both floc and snowflake yeast showed overall fitness increases in our experimental regime. However, floc yeast showed a significantly higher increase in fitness than snowflake yeast, increasing by an average of 15% as opposed to 7%, respectively (*Figure 3A* first panel: one-sample $t$-test, n=20, p<0.0001). Floc yeast showed a significant fitness advantage during growth (*Figure 3A*; mean increase = 24%, one-sample $t$-test, n=20 $t$=10.55, p<0.0001), but despite settling an average of 12-fold faster than their ancestor when grown in monoculture, were only slightly significantly more fit than their ancestor during settling selection (*Figure 3A*; mean increase = 3%, one-sample $t$-test, n=20, $t$=1.633, p=0.12). Snowflake yeast displayed the opposite behavior, possessing a 5.5% fitness benefit during growth (*Figure 3A*; one-sample $t$-test, n=20, $t$=8.374, p<0.0001) and a 28% fitness advantage over their ancestor during settling selection (*Figure 3A*; one-sample $t$-test, n=20, $t$=10.29, p<0.0001). Despite floc as an overall treatment group not having higher fitness during settling selection, isolates from 2/20 replicate populations did have detectably higher fitness than their ancestor when competing during settling selection ($F_{20,63}$=4.528, p<0.0001, multiple comparisons controlled by a Bonferroni correction with overall $\alpha$=0.05, the same two populations were identified using a Dunnett's test against an ancestor:ancestor control). In contrast, 12/20 snowflake populations significantly increased fitness during settling selection ($F_{20,63}$=4.89, *p*<0.0001, multiple comparisons controlled using a Dunnett test against ancestor:ancestor control). To determine why floc obtained such marginal fitness benefit during settling selection despite evolving such a large increase in their rate of flocculation, we measured the genetic structure of flocs formed by five 24-week isolates from across the range of settling speeds by calculating their assortment (*Yanni et al., 2019*), a scaled statistic of genotypic enrichment relative to what would be expected by chance, which ranges from –1 to 1 (representative images of flocs, compressed to a single-cell thickness, shown in *Figure 3C and D*). Multicellular adaptation requires a positive correlation between group phenotype and underlying cell-level genotype; without this, selection acting on groups cannot drive changes in allele frequency (*Clarke, 2014*; *Fletcher and Doebeli, 2009*; *Pepper and Smuts, 2002*). Genotypes that achieve high assortment should thus have greater potential for multicellular adaptation.

Overall assortment in floc was relatively low (mean of five randomly-selected strains from *Figure 3B* was 0.06), though there was significant among-strain variation (one-way ANOVA, $F_{4,10}$=4.159, p=0.008). Low assortment impedes the potential for selection acting on groups to drive changes in allele frequencies, explaining why evolved floc yeast obtained little fitness benefit from their remarkably improved sedimentation rates. In contrast, snowflake yeast canonically have an assortment of 1 (when competing two strains, every group is entirely clonal *Figure 2—figure supplement 3*; *Pentz et al., 2020*), allowing for selection acting on emergent group-level traits (i.e. settling speed) to act on underlying genetic mechanisms.

## Genomic analysis provides insight into evolutionary dynamics

We sequenced the genome of one isolate per 24-week population (40 isolates in total), representing about 700 generations of evolution (*Figure 4—figure supplement 1*). Floc yeast accumulated more mutations than snowflake yeast, an average of 5 vs 3 mutations per genotype, respectively (*Figure 4A*;

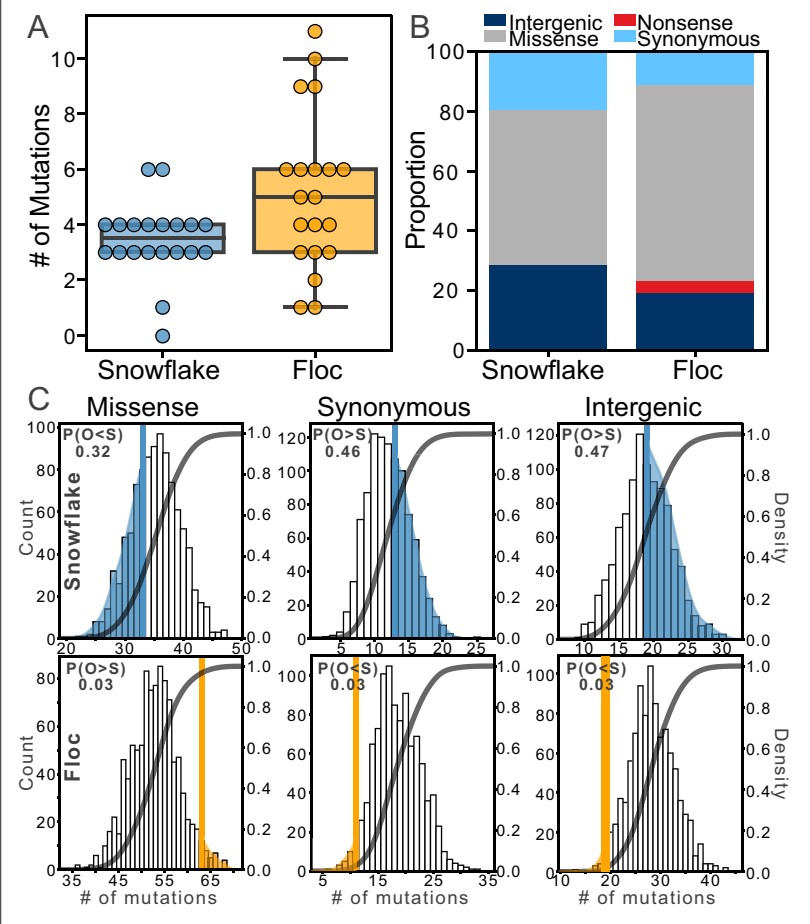

**Figure 4.** Examining mutations for signatures of selection. After 24 weeks of evolution, floc yeast accumulated more mutations on average than snowflake yeast (5 vs 3, respectively, **A**). We categorized these mutations into four broad classes (**B**), then compared the number of each type of mutation to expectations based on a simulation of neutral evolution (**C**). We cannot distinguish the overall pattern of mutations in snowflake yeast from neutral expectations. In contrast, floc yeast showed strong evidence of selection, with more missense mutations and fewer synonymous and intergenic mutations than expected. The number in the upper left-hand corner of each subplot reports the shaded area of each distribution, which is the proportion of simulated runs in that observed a result at least as extreme as the observed number of mutations of that type we identified, pooling across all 20 floc and snowflake genotypes.

The online version of this article includes the following figure supplement(s) for figure 4:

**Figure supplement 1.** Number of doublings per day in snowflake and floc yeast over the course of our 24-week selection experiment.

**Figure supplement 2.** Growth adaptation control for floc.

two-tailed *t*-test, t=2.49, n=40, p=0.017). All mutations are listed in *Supplementary file 1* (Snowflake) and *Supplementary file 2* (Floc). To develop insight into how each life cycle affected the potential for selection to act on mutations, we compared the frequency of different types of point mutations (which constituted the large majority of total mutations; *Figure 4B*) to those predicted by a null model of evolution lacking selection. Specifically, we generated a distribution of the expected frequency of different types of mutation (i.e. missense, nonsense, synonymous, or intergenic) in yeast by simulating 100,000 mutations using the Python package Mutation-Simulator (*Kühl et al., 2021*), then ran 1000 bootstrap simulations in which we sampled the actual number of mutations identified in both floc (104 mutations total) and snowflake (69 mutations total) populations. Then we compared the observed frequency of these four types of mutation in both floc and snowflake populations to this null distribution, which is the distribution of mutations expected without selection (i.e. under genetic drift).

The overall pattern of mutations observed in snowflake yeast were not distinguishable from drift, with the number of mutations in each of the four categories being near the mean of the expectation under selective neutrality (*Figure 4C*). In contrast, the mutations in floc yeast show a strong signature of selection. We observed more missense mutations than expected based on drift (there was a 3% chance of observing at least this many missense mutations in our null model). Similarly, we observed fewer synonymous mutations and fewer mutations in intergenic regions than expected based on our null model (in both cases, there was a 3% chance of observing this many mutations or fewer in our null model). Nonsense mutations occur infrequently, and do not provide a sufficiently large sample size for analysis. Because we did not have a prior expectation about the deviation from the null, each of the above tests should be treated as a two-sided test, meaning there is a 6% chance of observing a deviation in either direction at least as large as the one observed for the three categories of mutation described above, even when the null hypothesis of no selection is true. Taken together, our results show that selection was acting strongly on the mutations found in floc, but not snowflake yeast. The analysis above only examined the overall statistical properties of mutations, and should not be interpreted to mean that snowflake yeast did not undergo adaptive evolution. Indeed, we see that all evolved isolates taken from all 20 populations have evolved to form larger groups (*Figure 2A–E*, *Figure 2—figure supplement 1*) that settle faster (*Figure 1C*) and are much more fit (*Figure 3A*). Some of the mutations identified within snowflake yeast are putatively adaptive, such as missense mutations in the cell cycle (i.e. *ULP1, SLK19*; *Supplementary file 1*) or mutations affecting cellular morphology (i.e. *AYR1, ACM1, GSC2, CHC1, ARP7, HKR1*), which prior work in snowflake yeast has shown are a common mode of evolving larger group size (*Dahaj et al., 2021*; *Bozdag et al., 2023*; *Bozdag et al., 2021*; *Jacobeen et al., 2018*). We saw only a single *GAL2* mutant (in contrast to seven arising independently in floc, *Supplementary file 2*), which is consistent with less growth rate adaptation in snowflake yeast. Indeed, it is plausible that selection acting on multicellular traits also contributed to genetic drift through the hitchhiking of otherwise non-adaptive alleles (an outcome that has been seen previously in yeast selection experiments *Lang et al., 2013*).

Our results (*Figure 3*) suggest that faster settling in floc, in contrast to snowflake yeast, is only marginally adaptive. Yet if that is the case, why would floc convergently evolve to settle an average of 12 times faster than their ancestor (*Figure 1*)? We see two potential explanations, which are not mutually exclusive. The first is that low genetic assortment during flocculation reduces, but does not eliminate, the benefit of faster settling. Indeed, we found that 24 week evolved floc were an average of 10% more fit than their ancestor during settling selection, although this difference was not statistically significant for the treatment as a whole. Consistent with flocculation itself being under selection, we saw five parallel missense mutations in *FLO1*, and four in *MSN1*, a transcriptional activator of another major flocculin gene, *FLO11* (*Fichtner et al., 2007*; *van Dyk et al., 2005*; *Bayly et al., 2005*; *Supplementary file 2*). Greater statistical power during competitions may be required to detect the benefit of increased flocculation if it exists, however.

Alternatively, it may be that faster settling has evolved as a pleiotropic consequence of increased growth rate. In our experiments, mutations that increase growth rate by altering carbon metabolism may also increase flocculation efficiency as a side-effect, because *FLO1* expression is under the control of a *GAL1* promoter. We saw seven parallel missense mutations in *GAL2*, a transmembrane galactose permease, which may increase the amount of galactose entering the cell. As the main carbon source in the medium and promoter of *FLO1*, this may both increase growth rate and flocculation rate simultaneously. We conducted an additional experiment to control for this possibility, evolving five floc genotypes for eight weeks with daily selection for growth rate, but not settling selection, in the same growth medium containing galactose. These controls evolved a 20.7% increase in fitness during the growth phase, but evolved a 40% decrease in flocculation efficiency (*Figure 4—figure supplement 2*). We conclude that cell-level adaptation to growth on galactose does not drive increased flocculation as a pleiotropic side-effect. It is thus likely that the dramatically increased flocculation efficiency seen in our main experiment (*Figure 1B*) is an adaptation, albeit one that provides only a relatively small advantage (*Figure 3A*).

## Life cycles result in different genetic bottlenecks: modeling adaptation in aggregative and clonally-developing life cycles

To reconcile the results above, in which clonal multicellularity both facilitates multicellular adaptation yet experiences more genetic drift, we developed a theoretical model. Specifically, we consider the fate of a rare, beneficial mutant lineage that arises during the population growth phase. If the mutant lineage is to eventually fix in the population, it must survive settling selection, which means being in at least one group that gets selected. The probability that at least one mutant cell survives settling selection depends on the fraction of groups that contain at least one mutant cell.

As a point of comparison we begin by considering a simple population of equal-sized groups in which each group contains exactly one mutant cell. In this scenario if there are $k$ mutant cells then there are also $k$ groups with a mutant cell. Since all groups are the same size, we can model the process of selection as a statistical sampling procedure in which groups are sampled randomly without replacement. For the mutant lineage to go extinct none of the $k$ groups containing a mutant can be selected. The probability of such an event (call it $p_e$) is described by a hypergeometric distribution: $p_e = \prod_{1=0}^{fN} \frac{N-k-i}{N-i}$, where the total number of groups selected is expressed as a fraction f of a total population of $N$ groups. If $N$ is large, we can simplify this to $p_e \approx e^{-kf/(1-f)}$, which shows that the probability a mutant lineage goes extinct exponentially decays as $k$ increases.

The value of $k$ depends on how mutant cells are partitioned across groups. In the above case we assumed for simplicity that each group contained only a single mutant cell. Such a scenario could arise in a population without preferential assortment (i.e. an idealized form of aggregative development) if the proportion of mutant cells is small in comparison to the number of groups. In contrast, when groups are highly assorted (e.g. clonal development), the initially rare mutant and all of its descendants will be represented in a small number of groups, reducing $k$ by a factor that corresponds to mean group size. To get a sense of this effect, suppose a mutant arises at the beginning of the growth phase. The population then grows 100-fold, resulting in 100 mutant cells. Assuming the average size of a group is 50 cells, then if 1% of the population survives settling selection, the probability that the mutant lineage goes extinct in a population that develops clonally is 98%, while it is just 36.4% for a population developing aggregatively. To assess the robustness of our theoretical predictions, we used experimental data measuring the distribution of group sizes in floc and snowflake yeast and simulated different mutations arising in these populations. We considered mutations that either: 1. Alter cell growth by a factor $(1+s_c)$ where positive $s_c$ corresponds to a beneficial mutation, or 2. Improve the settling rate of that group by a factor of $s_g \cdot m$, where $m$ is the number of mutant cells. We assume the best scenario for a mutation to fix, that is, it arises at the very start of an exponential growth phase. We then used experimentally observed size distributions of groups to allocate the offspring of this original mutant cell. For simplicity, in the well-mixed aggregative life cycle, we randomly selected cells to be mutant, thereby causing larger groups to have more mutant cells (although keeping the proportion of mutants similar). For the clonal life cycle, we randomly selected groups, allocating mutant cells to it until we either ran out of cells, or had to choose a new group (biologically, these would be propagules of the first cluster, as they all descend from a common mutant ancestor). After allocating mutants, we simulated settling selection by probabilistically selecting groups, with survival weighted linearly by size. We iterated over five rounds of growth and size-dependent selection.

We recapitulate key dynamics from our experiments in this simple model. In the aggregative life cycle, selection readily acts on mutations that affect cellular growth rate (either favoring beneficial mutants or purging those that are deleterious; *Figure 5A*). In contrast, the clonal life cycle struggles to capture growth beneficial mutations- even mutants that doubled cellular growth rates went extinct ~40% of the time. When we examine mutations that increase group survival, we find that while they rarely go extinct in aggregative groups (*Figure 5B*), neither can they be efficiently selected upon (*Figure 5C&D*). In the clonal life cycle, even strongly group-beneficial mutants (e.g., increasing the probability of group survival 10-fold) go extinct far more often than not, as they are extremely susceptible to being lost in the first several rounds of the simulation when they are found in only a small number of groups. If they escape being lost by drift, however, they rapidly fix in the population (*Figure 5C&D*). Additionally, we show that these general results are robust to variation in the overall bottleneck size (*Figure 5—figure supplement 1*). The clonal life cycle thus allows selection to act, albeit inefficiently, on group-survival beneficial mutations, while the aggregative life cycle is strongly biased in favor of capturing growth-enhancing mutations.

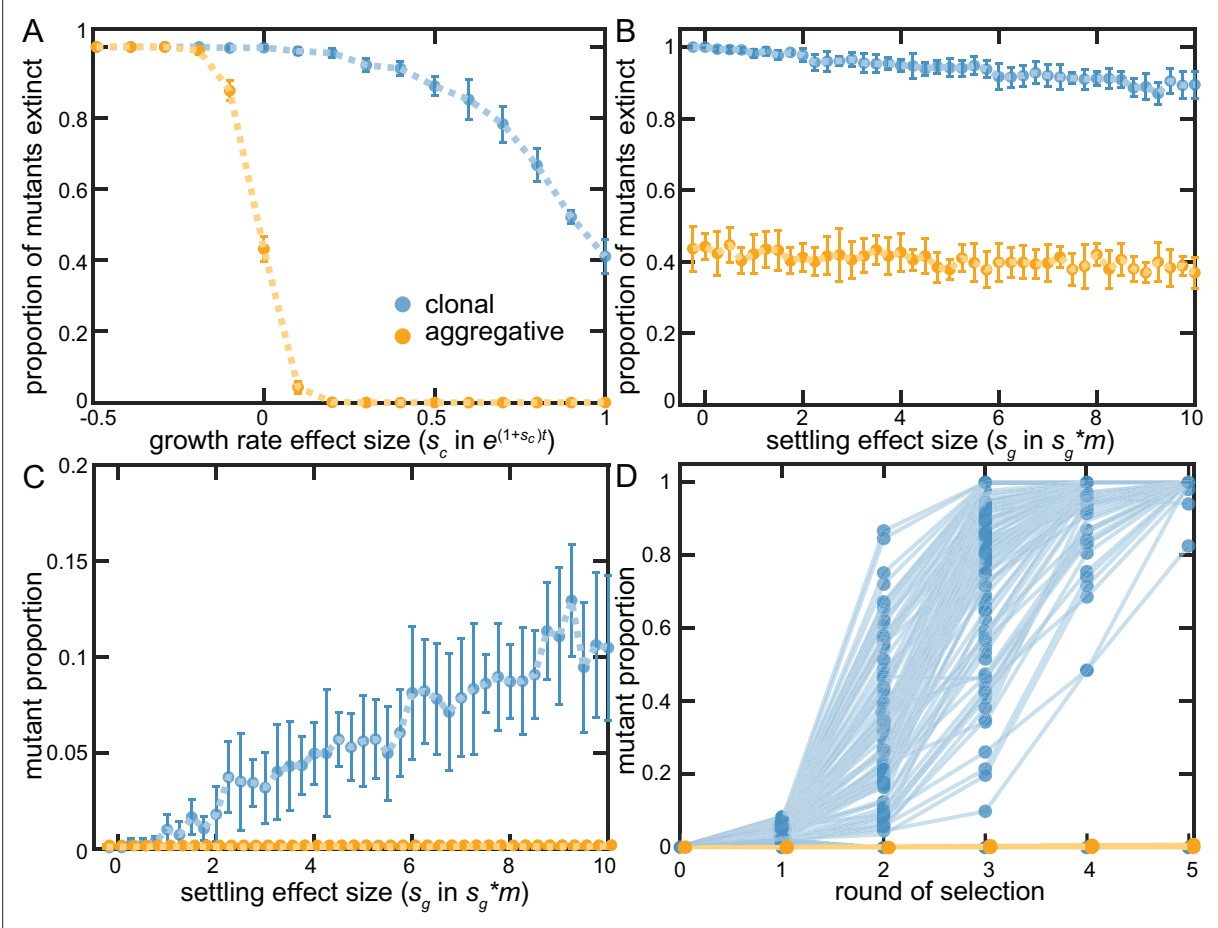

**Figure 5.** Examining clonal vs. aggregative development in a simple model. We examined the ability of organisms that develop clonally or aggregatively to fix mutations that increase growth rate (**A**) or group survival (**B–D**). Aggregative organisms readily fixed growth beneficial mutations (**A**), but were unable to act upon group-beneficial mutations (**B&C**). In contrast, clonally-developing organisms were capable of fixing mutations that improve both growth and the survival of their group, though they faced far more stochastic loss of each type of beneficial mutation than floc. **B-D** show that clonal development stochastically loses even strongly group beneficial mutants a majority of the time due to sampling error (in **D**, $s_g$ = 10), but when these mutations persist beyond the first two rounds of selection, they rapidly fix. Note that 88% of the beneficial mutants within the clonal lineages went extinct in **D**.

The online version of this article includes the following figure supplement(s) for figure 5:

**Figure supplement 1.** Fixation of beneficial mutations in floc and snowflake yeast under different bottleneck sizes.

## Discussion

During the evolutionary transition to multicellularity, groups of cells become Darwinian individuals, capable of reproducing and possessing heritable variation in multicellular traits that affect fitness (*Buss, 2014*; *Godfrey-Smith et al., 2013*; *Clarke, 2014*; *van Gestel and Tarnita, 2017*; *Libby and Rainey, 2013*; *Rainey and De Monte, 2014*). Using a simple yeast model system, in which we engineered either an aggregative life cycle or one in which groups develop clonally, we used experimental evolution to examine how developmental mode impacts this evolutionary transition in individuality. Our two strains each differ from the common ancestor by a single mutation (loss of *ACE2* for snowflake yeast, and gain of *FLO1* functionality for floc), highlighting how initially trivial differences in the mode of group formation may underpin divergent evolutionary trajectories. The clonal multicellular life cycle, but not aggregation, allowed for efficient multicellular adaptation in response to selection on group-level traits, suggesting that in these populations there was a shift in the level of Darwinian individuality.

While both clonally-developing snowflake yeast and aggregative floc evolved to settle faster in response to settling selection (forming larger, faster-settling clusters or flocs, respectively), only

snowflake yeast showed a shift in the balance of selection from cells to groups. When competing evolved isolates against their ancestor, we found that snowflake yeast improved their fitness mainly during settling selection (a phase of the culture cycle that primarily reflects group-level properties), not the growth phase of the experiment (a period of the culture cycle that mainly reflects changes in cellular growth rate). Floc yeast adapted in the opposite manner, gaining a majority of their fitness increase during the growth phase, with little detectable increase in fitness during settling selection. When floc cells aggregate, they form groups with a genetic composition resembling the population as a whole. This low genetic assortment diminishes the ability for group-level selection to drive evolutionary change, impeding the transition to multicellular individuality.

The differential pattern of mutations in floc and snowflake yeast, informed by a simple mathematical model, highlights another significant impact of the mechanism of group formation: the rate of genetic drift. In floc yeast, natural selection efficiently acted on novel mutations, with fewer synonymous and intergenic mutations than expected by chance, and more missense mutations than expected by chance. As our method detects a divergence in the proportion of each mutation type from neutrality, these results could either be the result of selection against synonymous and intergenic mutations, selection for missense mutations, or both. While synonymous mutations may be non-neutral (*Bailey et al., 2021*; *Shen et al., 2022*), the vast majority of mutations that fix during microbial experimental evolution are nonsynonymous (*Lang et al., 2013*; *Bailey et al., 2021*), as these are often subject to positive selection (*Good et al., 2017*; *Johnson et al., 2021*). When we query all putative loss of function mutations that arose in our experiment in a genome-wide deletion collection grown on galactose (*Giaever et al., 2002*), we find that 46% of the mutations in floc are predicted to increase growth rates on this carbon source, in contrast to just 35% in snowflake yeast. This is consistent with selection on mutations affecting cellular growth rate being more efficacious in floc than snowflake yeast.

Snowflake yeast, on the other hand, showed no statistical deviation from the expectations of neutrality. While initially surprising, this finding is consistent with group formation and selection playing a more central role in their evolution. Our model demonstrates that clonal development allows selection on group-level traits to be more efficacious, but simultaneously drives a much a higher overall rate of genetic drift, due to the higher probability that novel mutants will be lost during the group selection phase of the experiment. This result echoes prior theoretical work showing that the effective population size in a metapopulation is greatly reduced by strong among-deme selection, limited migration, and a small number of colonizing cells relative to carrying capacity (*Whitlock and Barton, 1997*).

Our model highlights how the severity of genetic bottlenecking due to clonal development scales with organismal size, with larger organisms exacerbating drift. Relative to the standards of most extant multicellular organisms (*Bell and Mooers, 1997*), snowflake yeast are small. As a result, the realized disparity between clonal and aggregative development in many organisms may be considerably greater than what was observed in this paper. It is not yet clear what impact this has on the evolution of complex multicellularity. Reducing the ability for selection to act efficiently on cell-level fitness (i.e. growth rate) may relax constraints on the evolution of novel group-level traits that come at a cost to growth (*Herron et al., 2009*). In addition, relaxed selection can facilitate a broader search of potential trait space, which may open up novel adaptive routes that would have been constrained by strong selection. This may be especially important during an evolutionary transition in individuality, as the nature of the organism is being fundamentally reshaped and genotypes are presumably far from potential fitness optima. Indeed, relaxed selection appears to have played a central role in the evolution of larger and more complex genomes within eukaryotes (*Lynch and Conery, 2003*), freeing these populations from the ruthless efficiency of strong purifying selection.

While floc and snowflake yeast differ starkly in evolutionary mode, our experiment was too short to examine long-term evolutionary consequences. Large scale change is possible in our system, but the evolution of macroscopic size appears to be much more challenging for snowflake yeast than in floc, requiring more innovation. Snowflake yeast must resolve fundamental biophysical challenges before they can form large groups, evolving larger size via three distinct biophysical mechanisms: reducing cellular density within clusters to limit cell-cell jamming, increasing the size and strength of the bonds connecting mother and daughter cells, and evolving branch entanglement so that group fracture requires breaking many cellular bonds, not just one (*Bozdag et al., 2023*). These innovations took 400–600 serial transfers to evolve, and resulted in snowflake yeast that were 20,000 times larger than

their ancestor, and 10,000 times more biomechanically tough. In contrast, evolving macroscopic size (*Figure 2F*) appears somewhat trivial for floc, likely due the relative ease and efficacy of increasing the strength of cell-cell adhesion.

There is an emerging consensus that clonal multicellular development provides the foundation upon which sustained multicellular adaptation may ultimately drive the evolution of dramatically increased multicellular complexity. This is supported by phylogenetic inference (*Fisher et al., 2020*; *Fisher et al., 2013*; *King, 2004*), first-principles theory (*Tarnita et al., 2013*; *Márquez-Zacarías et al., 2021*; *Ratcliff et al., 2017*; *van Gestel and Tarnita, 2017*; *Grosberg and Strathmann, 1998*; *Queller, 2000*), and now, experimental evolution. In this paper, we show that only clonally developing snowflake yeast exhibited a shift in Darwinian individuality (*Godfrey-Smith et al., 2013*; *Rose and Hammerschmidt, 2021*), such that multicellular groups were the primary unit of selection, with increased fitness arising mainly via the group-selection phase of the life cycle. Aggregative floc, in contrast, behaved as a social unicellular organism, adapting in both phases of the experiment, but low assortment limited the potential fitness benefits of selection acting on groups. Persistent low genetic assortment during group formation presents a challenge to this evolutionary transition, blunting the impact of group-level selection, enabling within-group genetic conflict, and ultimately inhibiting a shift in the level of Darwinian individuality from cells to groups.

And yet, aggregative multicellularity is clearly a successful life history strategy, evolving repeatedly in diverse lineages despite the issues described above. Our results thus raise a number of key questions for future research: how do lineages evolving aggregative multicellularity overcome the constraint of limited assortment? Is active kin recognition a pre-requisite to the evolution of multicellular individuality for aggregative life cycle, or can spatial structure generated by a patchy environment or viscous media like soil (*Yanni et al., 2019*) provide sufficiently high assortment for efficient, sustained multicellular adaptation? To this latter point, most lineages of aggregative multicellularity are terrestrial (*Bonner, 1998*; *Fisher et al., 2020*) - does this reflect the potential of for a highly structured environment to scaffold the origin of aggregative multicellularity (*Black et al., 2020*), or are there simply more ecological opportunities for a biphasic life cycle in terrestrial habitats? We still have only a rudimentary understanding of how key components of early multicellular life cycles (e.g. developmental mode, level of genetic assortment within groups, type and strength of selection on cellular and group-level traits, etc.) influence the evolution of multicellular complexity. Developing a robust, bottom-up understanding of multicellular evolution from first-principles theory will require the integration of multiple approaches- including mathematical modeling, fieldwork, and experiments incorporating both synthetic and naturally evolved multicellular organisms.

## Methods
### Strains and media

Strains used in this study are listed in *Table 1* and the construction of these is described in *Pentz et al., 2020*. Briefly, flocculant yeast were created by replacing the *URA3* open reading frame (ORF) with the *KAN-GAL1p::FLO1* cassette (*Smukalla et al., 2008*) and snowflake yeast were created by replacing the *ACE2* ORF with the drug marker, *KANMX*. These genotypes were created from the same homozygous diploid unicellular ancestor (*Saccharomyces cerevisiae* strain Y55, accession JRIF00000000), so these strains differ only in their mode of cluster formation. All experiments were performed in rich medium composed of a mix of glucose and galactose (YPGal +Dex; per liter; 18 g galactose, 2 g glucose, 20 g peptone, 10 g yeast extract), shaking at 250 rpm at 30 °C. These growth conditions yield clusters of similar size after 24 h of growth.

**Table 1.** Strains used in this study.

| Strain | Relevant Genotype | Reference |
|---|---|---|
| Snowflake | Δ*ace2*::HYGMX / Δ*ace2*::HYGMX | *Pentz et al., 2020* |
| Floc | Δ*ura3*::KAN-GAL1p::*FLO1* / Δ*ura3*::KAN-GAL1p::*FLO1* | *Pentz et al., 2020* |
| Snowflake-GFP | Δ*lys2*::*TEF2*p-yeGFP / Δ*lys2*::*TEF2*p-yeGFP | *Pentz et al., 2020* |
| Floc-GFP | Δ*lys2*::*TEF2*p-yeGFP / Δ*lys2*::*TEF2*p-yeGFP | *Pentz et al., 2020* |

## Experimental evolution

Twenty replicate populations of both snowflake and floc yeast were initiated into 10 mL of YPGal +Dex from a single clone and grown overnight. Every 24 h, each population (40 populations total) was subjected to daily selection for settling for 5 min on the bench as described in *Ratcliff et al., 2012*. Briefly, 1.5 mL of each overnight culture was placed into a 2 mL microcentrifuge tube, left to settle on the bench for 5 min, after which the top 1.4 mL of culture was discarded. The remaining 100 µL of the pellet was transferred to a fresh tube containing 10 mL of culture media for the next round of growth and selection. Our selection regime yielded ~4–5 generations per day for both snowflake and floc yeast (*Figure 4—figure supplement 1*). Every 7 days, whole populations were cryogenically stored at –80 °C. As a control, five populations of snowflake and floc yeast were evolved without settling selection for eight weeks. Specifically, five replicates of both snowflake and floc yeast were initiated as described above. Every 24 hr, each population (10 populations total) was briefly vortexed, then 100 µL was transferred to 10 mL of fresh medium. In the absence of settling selection, this would result in 6.6 generations of growth per day. The number of generations realized per transfer was lower than this upper ceiling (*Figure 4—figure supplement 1*), because groups of cells surviving settling selection allow more biomass to be transferred. Whole populations were cryogenically stored at –80 °C every 7 days.

## Measuring settling rate

To explore the dynamics of multicellular adaptation in floc and snowflake yeast populations, we measured the settling rate of each population every 7 days over the 24-week experiment. We thawed cryogenically-stored whole populations and subsequently inoculated 100 µL into 10 mL of YPGal +Dex and grew them for 24 hr. Then, 100 µL of overnight cultures was inoculated into fresh YPGal +Dex media and grown for an additional 24 hr. We measured the settling rate of populations as described in *Pentz et al., 2020*. Briefly, high-speed high-resolution videos of yeast populations settling in back-illuminated cuvettes were recorded using a Sony a7RII and 90 mm macro lens (24 fps, 3840x2,160 pixels). Then, custom scripts were used to determine the rate of yeast biomass displacement, or settling rate, based on changes in pixel densities over settling time (*Pentz et al., 2020*).

## Phenotypic assessment of evolved populations

Size distributions of evolved snowflake yeast were obtained via flow cytometry on a CyFlow Space flow cytometer using the forward scatter (FSC) channel as a proxy for cluster size. To account for the fact that larger clusters contain more biomass, we calculated a biomass-weighted mean size for each population. To do so, we developed a Python script that divides the distribution of cluster size into 100 bins, then determines the proportion of the population's total biomass that is found in each bin (sum of the FSC values for each cluster within that size range). This gives us a new distribution of the population's biomass across the range of cluster sizes. A sensitivity analysis showed that this algorithm is robust to bin size (*Figure 2—figure supplement 4*) and the calculated means of the biomass-scaled distribution (reported in *Figure 2A*) are not affected by the bin size (*Figure 2—figure supplement 4*).

Flocculation efficiency was measured using the first frame from the timelapse videos used to measure settling rate in the week-24 evolved floc populations, as well as the ancestor (*Pentz et al., 2020*). The population is well-mixed in this frame, so floc aggregates will result in optically dense regions while the planktonic culture (single cells that are not in flocs) will be less optically dense. Thus, higher flocculation efficiency will result in a higher variance in the pixel density between floc aggregates and planktonic cells. A custom Python script was used to calculate the maximum variance in pixel density on four biological replicates for each population. A histogram of pixel brightness of the two floc populations with highest flocculation efficiencies (populations F7 and F18, see *Figure 2G*) showed that pixel opacity does not saturate at either high or low pixel values representing cells in dense flocs or un-flocculated cells (*Figure 2—figure supplement 2*), respectively, showing that this approach broad enough dynamic range to determine flocculation efficiency.

## Fitness competitions

To determine the fitness of evolved populations, a representative genotype was isolated from each population at the end of the experiment (24 weeks) by three rounds of single-colony selection on YPGal +Dex agarose plates (YPGal +Dex with 15 g/L agarose). Our selection regime is characterized

by fluctuating periods of selection for growth and selection for rapid settling, or large size (**Pentz et al., 2020**; **Ratcliff et al., 2012**). Thus, it is important to measure fitness in both of these important life history traits. To do so, we quantified the fitness of evolved isolates relative to their ancestor over one round of growth and one round of settling selection. Specifically, to initiate competitions, we inoculated 10 mL cultures of YPGal +Dex with isolates from each population as well as a GFP-marked ancestor and grown for 24 hr. Then, we mixed each of the evolved isolates in equal volumes with its marked ancestor (floc or snowflake), and 100 μL of this mixture was inoculated into 10 mL of YPGal +Dex to start competitions. For snowflake competitions, whole cluster counts of the GFP-tagged ancestor and evolved isolate were obtained via flow cytometry using a CyFlow Space flow cytometer where GFP and non-GFP clusters can be distinguished using the FL1 fluorescence channel. For floc competitions, flocs were deflocculated using 50 mM EDTA (pH 7) prior to running on the flow cytometer, and cell counts of the GFP-tagged ancestor and evolved isolate were obtained similar to snowflake competitions. Counts were obtained at time 0 and after 24 hr of growth to determine the fitness of the evolved isolate over one period of growth. To measure fitness over one round of settling selection, 2 mL of the overnight mixed culture was aliquoted into a microcentrifuge tube, and 500 μL was used to determine pre-selection counts. The remaining 1.5 mL was used to perform one round of settling selection (5 min on the bench), after which the top 1.4 mL was discarded. The remaining bottom 100 μL was used to determine post-selection counts. In all competitions, relative fitness was calculated using the ratio of Malthusian growth parameters (**Lenski et al., 1991**). Relative fitness was normalized to the fitness of the ancestral strain for each environment.

Fitness was also measured for the five control populations of floc yeast relative to their ancestor over three transfers. Competitions were initiated as described above. Every 24 hr for 3 days, 100 μL was transferred to fresh medium. Counts at the beginning and end of the competition were obtained using flow cytometry as described above, and the relative fitness was calculated using the ratio of Malthusian growth parameters (**Conlin and Ratcliff, 2016**).

## Floc assortment measurements

To calculate assortment, we co-cultured GFP-tagged evolved strains with an RFP-tagged ancestor. Strains were grown overnight at 30 °C in 10 mL of YP +1.8% galactose +0.2% dextrose. The next day, cultures were mixed by vortexing and a 1 mL sample was deflocculated by centrifuging and resuspending in 100 mM EDTA. To remove EDTA, which inhibits growth, strains were again centrifuged and resuspended in YP +1.8% galactose +0.2% dextrose. From this, 100 μL of each strain to be co-cultured were added to 10 mL of the same media to grow overnight at 30 °C. For imaging the next day, co-cultures were vortexed for 10 s, and 1.5 mL of culture was added to each of two tubes. One tube was for measuring the baseline population frequency of the two strains, and EDTA was added to a final concentration of 100 mM. The other tube for each co-culture was for settling selection, and was left on the bench for 5 min, then all but the bottom 100 uL was removed. The remaining 100 μL were deflocculated by adding EDTA to a final concentration of 200 mM. Both samples were concentrated by centrifugation, and a small sample was imaged with a 20 X microscope objective. Three images were taken for each sample. Cells in the red and green channels were counted using Otsu thresholding and watershedding in FIJI. Assortment was calculated using the following equation that controls for population frequency, where $f_{set}$ and $f_{pop}$ are the frequency of the evolved strain after settling selection and in the general population, respectively:

$$Assortment = \frac{f_{set} - f_{pop}}{1 - f_{pop}}$$

## Genomic DNA preparation

To determine the genetic basis of observed fitness differences, we performed whole-genome sequencing of 24-week evolved isolates and the starting ancestral genotypes. Yeast strains were streaked out for single colonies from –80 °C glycerol stocks. Single colonies were grown overnight in 10 mL YPGal +Dex and genomic DNA was isolated from 1 mL aliquots using the VWR Life Science Yeast Genomic DNA Purification Kit (VWR 89492–616, https://us.vwr.com/).

## Whole-genome sequencing

DNA libraries were prepared using the NEBNext Ultra II FS DNA Library Prep Kit for Illumina (https://www.neb.com/) and were sequenced on an Illumina HiSeq 2500. Paired-end 150 bp reads were used

for all samples. Mean coverage across the genome was 200 X for evolved isolate DNA and 50 X for ancestor DNA.

## Sequencing analysis

DNA sequences were quality trimmed using Trimmomatic (*Bolger et al., 2014*) and then aligned to the S288C reference genome R64-2-1 using the Burrows-Wheeler Aligner (*Li and Durbin, 2010*). Duplicates were marked using SAMBLASTER (*Faust and Hall, 2014*) then converted to a BAM file, then sorted and indexed. Variants were called using the Genome Analysis Tool Kit (GATK) Haplotype-Caller (*McKenna et al., 2010*). SNPs and INDELs were first filtered based on read depth and quality using vcffilter (https://github.com/vcflib/vcflib). Variants were removed with a read depth less than 10 and a quality score less than 20. Then, bcftools isec (https://github.com/samtools/bcftools, v1.18, RRID:SCR_002105; *Danecek et al., 2021*) was used to filter out variants shared between the ancestor and evolved isolates, accounting for variants called due to aligning *S. cerevisiae* strain Y55 used in our experiments to the S288C reference genome. Finally, bcftools isec was used again to identify unique variants for each evolved isolate. Variants were manually validated using the Integrated Genomics Viewer (*Robinson et al., 2011*). Final validated variants were pooled and annotated using SnpEff (*Cingolani et al., 2012*).

Next, we performed a bootstrap analysis to compare the classes of variants called experimentally to a randomly generated sample of SNPs to determine if different mutational classes are over- or underrepresented in our experimental populations. First, we used Mutation-Simulator (*Kühl et al., 2021*) to generate a null distribution of 100,000 random SNPs from across the *S. cerevisiae* S288C genome and annotated using SnpEff (*Cingolani et al., 2012*). Then, a custom Python script was used to perform a bootstrap analysis by first generating a random subsample of SNPs from the null distribution. The quantity of SNPs subsampled was equal to the pooled number of mutants seen experimentally for either snowflake or floc yeast (69 or 104 mutations, respectively). Next, we compared the number of SNPs in four mutational classes (missense, nonsense, synonymous/silent, and upstream gene variant) generated experimentally or simulated. We performed the bootstrap analysis 1000 times each for snowflake and floc yeast. Histograms for the number of simulated SNPs generated for different mutational classes in the bootstrap analysis can be seen in *Figure 4C* (experimental number shown as vertical blue and orange lines for snowflake and floc yeast, respectively). Finally, we determined the proportion (P) of runs where the # observed mutations > # simulated mutations and the P # observed mutations <simulated mutations (see *Figure 4C*). The code for the Python script is available at GitHub (https://github.com/gabe-dubose/emus; copy archived at *DuBose, 2023*).

## Mathematical modeling

We consider the survival and fixation of mutant lineages by distinguishing between two phases of the experiment between transfers: the population growth phase and the settling selection phase. During the population growth phase, we assume that both the mutant and ancestral lineages reproduce exponentially until the total population increases by a factor of 100, that is they reach the carrying capacity. So if the initial population is $I$ and there are $m$ mutants, we assume the population grows via $(I - m) e^{\lambda t} + me^{\lambda(1+s_c)t}$ , where the sign of $s_c$ determines whether the mutation is beneficial ($s_c$ >0) or deleterious ($s_c$ <0). We compute the number of mutants at carrying capacity and then place them in groups, according to whether we are simulating clonal or aggregative development. In the case of clonal development, we note that because of the branching pattern of snowflake yeast there can only be a maximum of one group that is mixed with mutant and ancestral lineages *Ratcliff et al., 2015*; *Libby et al., 2014*; all other groups with mutant cells are homogeneously mutant. We can simulate the distribution of mutants in groups of varying sizes by either tuning computational models of populations of snowflake yeast to fit experimental data or, instead, by directly using experimental data of group size distributions. We use the latter because it can easily be modified to consider aggregative development. Thus, for clonal development, we randomly select groups to place mutants. If there will be more mutant cells generated during growth (in this scenario, we are generating 100 mutant cells during the growth phase) than the size of the group, then the entire group is filled with mutant cells and a new group is selected to receive mutants, and so on until all mutant cells are allocated. In the case of aggregative development, we also use an experimentally derived distributions of group sizes but we select groups randomly weighted by their size to place individual mutant cells.

We simulate the settling selection phase of the experiment by randomly selecting groups weighted linearly by their size. For mutations that alter the survival of groups we assume that each mutant contributes an additional factor $s_g$ to the size of the group, that is the weight of a group is its size plus $s_g*m$, or equivalently if there are $n$ ancestral cells the weight is $n+(1+s_g)*m$. We then select groups randomly without replacement according to this weight until we have selected 1/100th of the population. If the number of cells exceeds 1/100th of the population we simply rescale the selected number of cells to fit. Following selection, we compute the number of surviving mutants and, if they exceed 0, we return to the growth phase. We iterate this process five times.

## Acknowledgements

This work was supported by NSF grant DEB-1845363 and a Packard Fellowship for Science and Engineering to WCR. KM was supported by NIH T32 grant T32GM142616. We thank Jordi van Gestel and the Ratcliff lab for constructive comments.

## Additional information

### Funding

| Funder | Grant reference number | Author |
| --- | --- | --- |
| National Science Foundation | DEB-1845363 | Jennifer T Pentz<br>James G DuBose<br>Peter L Conlin<br>Emma Reinhardt<br>William C Ratcliff<br>Kathryn MacGillivray |
| National Institutes of Health | T32GM142616 | Kathryn MacGillivray |
| Packard Foundation | Packard Fellowship for Science | Jennifer T Pentz<br>James G DuBose<br>Peter L Conlin<br>Emma Reinhardt<br>William C Ratcliff<br>Kathryn MacGillivray |
| Swedish Research Council | | Eric Libby |

The funders had no role in study design, data collection and interpretation, or the decision to submit the work for publication.

### Author contributions

Jennifer T Pentz, Conceptualization, Data curation, Formal analysis, Supervision, Investigation, Visualization, Writing - original draft, Writing – review and editing; Kathryn MacGillivray, Emma Reinhardt, Investigation, Writing – review and editing; James G DuBose, Data curation, Software, Formal analysis, Methodology, Writing – review and editing; Peter L Conlin, Conceptualization, Writing – review and editing; Eric Libby, Conceptualization, Software, Formal analysis, Methodology, Writing – review and editing; William C Ratcliff, Conceptualization, Supervision, Funding acquisition, Methodology, Project administration, Writing – review and editing

### Author ORCIDs

Jennifer T Pentz http://orcid.org/0000-0002-8186-3005
Peter L Conlin http://orcid.org/0000-0002-2793-7624
William C Ratcliff http://orcid.org/0000-0002-6837-8355

### Decision letter and Author response

Decision letter https://doi.org/10.7554/eLife.84336.sa1
Author response https://doi.org/10.7554/eLife.84336.sa2

## Additional files

### Supplementary files
- Supplementary file 1. List of mutations found in evolved snowflake yeast isolates.
- Supplementary file 2. List of mutations found in evolved floc yeast isolates.
- MDAR checklist

### Data availability
Complete sequencing data for all clones is available and NCBI BioProject PRJNA907014. The source code for the bootstrap analysis to compare mutational frequencies (Figure 4C) is available at https://github.com/gabe-dubose/emus (copy archived at *DuBose, 2023*).

The following dataset was generated:

| Author(s) | Year | Dataset title | Dataset URL | Database and Identifier |
|---|---|---|---|---|
| Pentz JT, MacGillivray K, DuBose JG, Conlin PL, Reinhardt E, Libby E, Ratcliff WC | 2022 | *Saccharomyces cerevisiae* strain:Y55 (baker's yeast) | https://www.ncbi.nlm.nih.gov/bioproject/?term=PRJNA907014 | NCBI BioProject, PRJNA907014 |

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
