## [Editor Report]

This study provides fundamental insight into the evolution of multicellularity by experimentally demonstrating that yeast strains that form clonal groups evolve stronger group traits than ones that aggregate into non-clonal groups. Through modeling and analysis of the mutations accumulated during the experiment, this paper provides compelling evidence in support of clonal development favoring selection at the group rather than the cellular level.

---

## [Decision Letter]

**Decision letter after peer review:**

[Editors’ note: the authors submitted for reconsideration following the decision after peer review. What follows is the decision letter after the first round of review.]

Thank you for submitting the paper "Evolutionary consequences of nascent multicellular life cycles" for consideration by *eLife*. Your article has been reviewed by 3 peer reviewers, including Sara Mitri as the Reviewing Editor and Reviewer #1, and the evaluation has been overseen by a Senior Editor. The following individual involved in the review of your submission has agreed to reveal their identity: María Rebolleda-Gómez (Reviewer #3).

Comments to the Authors:

First, I would like to sincerely apologize for how long it has taken to get back to you. We are also sorry to say that, after consultation with the reviewers, we have decided that this work will not be considered further for publication by *eLife*.

Even though two reviewers were very positive about the paper, one maintained that there were flaws with the experimental design. The first issue is that previous work has already demonstrated that selection for settling selects for Snowflake, not Floc, which makes it unsurprising that Snowflake is fitter than Floc under these same selective conditions. The second main criticism was regarding the way in which the two strains were created: the use of the GAL1 promoter may disadvantage the FLO1 strain such that the observed results are not due to inherent differences between aggregative and clonal multicellularity but are instead due to differences in strain background. We hope that the reviews, in addition to the lengthy discussion, which I have appended below, will be useful for you to revise the paper and resubmit it elsewhere.

*Reviewer #1 (Recommendations for the authors):*

I was very impressed by the experimental design and the scale of the experiments. My recommendations are all minor and/or because of interest, rather than being required for acceptance.

First, I found it interesting that the two strains differ only by 2 mutations. This means that the two potential evolutionary paths are actually very close together in origin. Does this have any significance for the evolution of multicellularity in yeast or is the selection regime too artificial to draw conclusions about these mutants in a natural environment? It might be interesting to add a few sentences on this to the Discussion section.

Another general comment I had was whether sedimentation is so obviously a group trait (L. 190-191). Although I am not so familiar with the literature, I know that this approach has been used for many years now, but is it that well-accepted? For example, the hydrodynamic profile was mentioned as something that could enhance sedimentation but sounds like an individual trait. This assumption is key to the whole conclusion of the paper, otherwise, it would not be justified to call it an evolutionary path towards multicellularity. In this sense, I find a bit strong in L. 460 to call selection for sedimentation "a shift in biological individuality".

Even though there is a section entitled "bottlenecks in the life cycle", bottleneck size is not being varied. Given that one of the main hypotheses of the model is that a strong bottleneck is what results in drift, it would be interesting to run a few more simulations to show that an increased bottleneck would lead to the stronger selection. The result is of course already known from the literature, but it would be good to verify that this occurs here.

A final point that could be discussed a bit more was the choice to pick single clones from each line. Is it possible that the single isolates you picked from each population may be cheaters, in particular in the floc lines? I understand that using 20 lines for each treatment protects to some extent against this, but it would be good to think about what this means statistically (e.g. if the majority of the population in an aggregate are cheaters, if there is some frequency-dependent selection, then in most lines you would miss the cooperators).

*Reviewer #2 (Recommendations for the authors):*

This study has a few fundamental flaws that substantially limit what can be concluded.

1) The logic and interpretation of the selection and fitness experiments are circular.

1a) The logic of the experiment is circular. As demonstrated previously by the authors, settling/centrifugation-based selection overwhelmingly selects Snowflake, not Floc.

However, the key points of this paper, is that using this selection regimen, they find that Snowflake is fitter than Floc under these selective conditions. Thus, the authors use a roundabout way to demonstrate that this selective regimen selects for Snowflake, not Floc, a fact already demonstrated.

1b) Related to point #1a, the second issue not considered is developmental constraints. For reasons that are unclear, Floc- yeast overwhelming seems to be selected for Snowflake. Thus, the cellular arrangements that promote this morphology are part of the evolutionary fitness valley that this strain is in that is highly optimal for this selective regimen. Indeed, the authors describe this in excellent detail in the discussion (that cellular connections matter for evolving Snowflake vs. Floc). Likewise, Floc in this regimen selects for stronger Floc, which is also in a fitness valley of its own. The selection regimen pushes these further into their fitness valleys that are sub-optimal under these selective conditions. Thus, the authors are not comparing the fitness of cellular-based undifferentiated multicellular versus aggregative undifferentiated multicellularity, they instead are measuring the fitness differences of the valleys these selected strains are in (and pushed further into), which again, as in point #1a overwhelmingly selects for Snowflake, not Floc. Thus, comparing an evolved strain optimized for selective conditions versus an evolved strain sub-optimal to the selective conditions.

1c) The key missing experiment is to reverse the selection. I.e. put Snowflake and Floc into a selective regimen that selects for Floc and asks which strain is fitter. However, should Floc be fitter in these conditions, it would demonstrate the null hypothesis here, which is that "you get what you select for". Without doing the experiment itself, however, interpreting the experiment as presented ends up being incomplete. This doesn't even get into more complicated selective regimens with something like predators and mouth/feeding size limits.

1d) One of the key points that do seem evident is that different modes of selection for the undifferentiated multicellular act at different levels (cell vs group). However, this is largely known as different developmental mechanisms of multicellularity are known to cell vs. group level selected. This experiment does demonstrate this point de novo, though the results are not unexpected.

2) This experiment, due to point #1, lacks generalizability. Modeling helps with generalizability, but I think empirical data is critical and easily obtainable here.

2a) Part of the way to address this would be by addressing point 1 above.

2b) The most logical experiment would be to repeat these experiments in a related system "multicellular" system under different selective conditions and ask which morphology is fitter. Some of the authors have previously selected multicellular *Chlamydomonas*. Or transitioning the experiments to *Dictyostelium* and selecting at different stages of its life cycle. One or both of these would be highly powerful in generalizing the potential findings.

3) What happens when selection is relaxed?

3a) One of the claims is that resources during growth vs. cell division are where selection is operating, but what happens when this is relaxed? Loss of the undifferentiated multicellularity at different rates when selection is relaxed would be highly informative as to which strain is fitter.

3b) One of the key things would be that in many organisms, motility will play a very large role in selection. As most unicellular Eukaryotes are motile (and their multicellular relatives – e.g. Chlamy and Dicty above), thus the fitness of group vs. individual may be substantially different when motility is taken into account.

3c) One potential aspect of this is the balance of epigenetic vs. genetic changes given the results of the genome re-sequencing analysis. I.e. group level trains may be selected at the genetic level, while epigenetic traits may be selected at the cellular level or vice-versa thereby complicating the conclusion of this work. The authors do indeed directly state this in lines 444-445 "While floc and snowflake yeast differ starry in evolutionary mode, our experiment was too short to examine the long-term evolutionary consequences". Which, if these facts are unknown, makes interpretation of the findings in this paper difficult to place especially given the nature of only examining one selective condition.

*Reviewer #3 (Recommendations for the authors):*

L. 92: Given that you do not explain yet that you are selecting for sinking, this sentence comes a little out of the blue (there are many other possible drivers of multicellularity). And given the start of the next paragraph, maybe it makes sense to specify that size selection is performed through settling.

L. 147 How is this affected by cell density – I understand that you are using the coefficient of variation to standardize over different densities, but I wonder if your pixel opacity saturates at a certain density (this is not a problem in comparing your ancestor to evolved because the differences are very clear, but it could be a problem between similar looking flocculation efficiencies). I wonder if instead, you could normalize pixel heterogeneity by cell number in some way.

Figure 2. – Change the ancestor reference line to have smaller dashes, or make it a different color. Dashes are the same width as the error bars and very similar to the width of the bars in the bar plot, making it visually harder to read the plot.

Line 534 – How robust is the algorithm to bin size?

It is not clear to me how competitions of snowflakes were performed – cell counts on a flow-cytometer? How did you perform the counts?

Line 328 – explain better why this is the case

This is beside the point, but I found interesting the last sentence… how can one get at first principles from fieldwork? Fieldwork helps us constrain the parameter space to what happens in nature, but I do not know if it is possible to get first principles when many things are all varying at once.

---

## [Author Response]

[Editors’ note: The authors appealed the original decision. What follows is the authors’ response to the first round of review.]

Comments to the Authors:First, I would like to sincerely apologize for how long it has taken to get back to you. We are also sorry to say that, after consultation with the reviewers, we have decided that this work will not be considered further for publication by eLife.Even though two reviewers were very positive about the paper, one maintained that there were flaws with the experimental design. The first issue is that previous work has already demonstrated that selection for settling selects for Snowflake, not Floc, which makes it unsurprising that Snowflake is fitter than Floc under these same selective conditions. The second main criticism was regarding the way in which the two strains were created: the use of the GAL1 promoter may disadvantage the FLO1 strain such that the observed results are not due to inherent differences between aggregative and clonal multicellularity but are instead due to differences in strain background. We hope that the reviews, in addition to the lengthy discussion, which I have appended below, will be useful for you to revise the paper and resubmit it elsewhere.Reviewer #1 (Recommendations for the authors):I was very impressed by the experimental design and the scale of the experiments. My recommendations are all minor and/or because of interest, rather than being required for acceptance.First, I found it interesting that the two strains differ only by 2 mutations. This means that the two potential evolutionary paths are actually very close together in origin. Does this have any significance for the evolution of multicellularity in yeast or is the selection regime too artificial to draw conclusions about these mutants in a natural environment? It might be interesting to add a few sentences on this to the Discussion section.

This is a good suggestion. We totally agree- these two paths, aggregation vs clonal development, are probably fairly close in origin in yeast. We explored just two mutational paths to get there, but there are many more (i.e., there’s a whole family of different flocculation genes, and mutations in many cell cycle genes result in snowflake-like clusters). However, it’s hard to say too much about the evolution of multicellularity in yeast in nature at this point- there’s still so much to learn about basic yeast ecology and their environmentally-dependent life cycles.

However, we think that your observation, that initially trivial differences in the mode of group formation can initiate divergent evolutionary dynamics that result in quite different long-term outcomes, is a general feature of clonal development vs. aggregation. The mechanistic underpinnings of either mode of group formation are likely relatively simple at first (e.g., its not hard to evolve to stick to other cells, forming aggregates, or remain attached after mitosis, forming clonal groups), even if the evolutionary dynamics that result from each mode of group formation are quite different. We added in a sentence to the discussion stating this explicitly:

Our two strains each differ from the common ancestor by a single mutation (loss of *ACE2* for snowflake yeast, and gain of *FLO1* functionality for floc), highlighting how initially trivial differences in the mode of group formation may underpin divergent evolutionary trajectories.

Another general comment I had was whether sedimentation is so obviously a group trait (L. 190-191). Although I am not so familiar with the literature, I know that this approach has been used for many years now, but is it that well-accepted? For example, the hydrodynamic profile was mentioned as something that could enhance sedimentation but sounds like an individual trait. This assumption is key to the whole conclusion of the paper, otherwise, it would not be justified to call it an evolutionary path towards multicellularity. In this sense, I find a bit strong in L. 460 to call selection for sedimentation "a shift in biological individuality".

This is a good point, and we don’t want to over-claim here. We are specifically referring to a type of biological individuality called “Darwinian individuality” by Peter Godfrey-Smith, which has been used quite elegantly as a framework for evolutionary transitions in individuality (see, for example, Rose and Hammerschmidt, 2021, Frontiers in Ecology and Evolution). Settling selection does indeed select directly on a group-level trait (speed of sedimentation), which can depend either additively or non-additively on underlying cell-level traits (see Bozdag 2023 Nature for a discussion of a non-additive case during longtermly evolution). However, it is important to clarify that it is not the selection on groups which indicates the origin of multicellular Darwinian individuality, but rather the capacity for the lineage to gain adaptations at that level via Darwinian evolution in response to this selection. While both snowflake and floc lineages underwent group-level selection during sedimentation, only snowflake yeast gained novel multicellular adaptations that increased fitness.

We clarified this in the paper by replacing “biological” individuality with “Darwinian”, and cite key papers on Darwinian individuality by Rose and Hammerschmidt, as well as Peter Godfrey-Smith.

The text was revised text is at line 475 and reads:

“In this paper, we show that only clonally-developing snowflake yeast exhibited a shift in Darwinian individuality^21,41^, such that multicellular groups were the primary unit of selection, with increased fitness arising mainly via the group-selection phase of the life cycle.”

Even though there is a section entitled "bottlenecks in the life cycle", bottleneck size is not being varied. Given that one of the main hypotheses of the model is that a strong bottleneck is what results in drift, it would be interesting to run a few more simulations to show that an increased bottleneck would lead to the stronger selection. The result is of course already known from the literature, but it would be good to verify that this occurs here.

It is true that we do not alter bottleneck size in the section entitled “bottlenecks in the life cycle”. Here the bottlenecks occur from the distribution of mutants across groups not necessarily the size by which the population contracts. Thus, clonal development experiences a stronger bottleneck than aggregation because its cells are in fewer groups. From this perspective, clonal development imposes the strongest bottleneck in terms of assortment and aggregation imposes the weakest—they are at opposite ends of the spectrum.

Based on the reviewer’s comments it sounds like they are interested in the extent to which the population contracts from settling selection, which does impose a bottleneck. However, the model in this section of the paper uses an assumption that the population size does not experience a net change in size over a round of growth and settling selection. So if the population expands by a factor of *X* in growth, it will decrease by a factor of 1/*X* in settling selection. Based on our analytical model the probability a neutral mutant goes extinct is *e*^–*kf*^/(1–*f*), where *k* is the number of mutants and *f* is the fraction selected (which is 1/*X*). We note that if we start the growth round with 1 mutant there will be *X* afterwards which means *k*=*X*. Thus the (*kf*) term is *X* × 1x = 1which means that the probability a mutant goes extinct is simply *e*^–*kf*/(1–*f*)^ which equals *e*^–*X*/(*X*–1)^, which is basically 1e for *X* > 25 =1 (the error in the approximation is about 4% and decreases as *X* increases). Increasing *X* which increases both the factor by which the population expands and the size of the bottleneck has a negligible effect on the probability that a mutant goes extinct. Thus, the bottleneck size in this section does not act like the reviewer’s expectation, but this is only because it also describes the extent to which the population expands in the growth phase.

That said, the factor *X* (which is both the bottleneck size and the extent to which the population expands) does have subtle effects on the graphs in Figure 5. We reran the simulations for *X*=50 and *X*=1000 (compared to *X*=100 in the original). We include them here along with the added Figure caption. One notable change is that the proportion of *sg* mutants that go extinct in aggregative life cycles decreases for *X*=1000 as compared to *X*=100 (see Figure 5B and 5C and J and K below) due to the odds of having multiple mutants in the same group increasing. This can greatly increase the probability that their group will survive. This same effect causes *sg* mutants to spread faster in both life cycles (see Figure 5D and Figure 5—figure supplement 1).

In addition, to make this clearer, we have change the title of this Results section to (line 329):

“Life cycles result in different genetic bottlenecks: modeling adaptation in aggregative and clonally developing life cycles”.

A final point that could be discussed a bit more was the choice to pick single clones from each line. Is it possible that the single isolates you picked from each population may be cheaters, in particular in the floc lines? I understand that using 20 lines for each treatment protects to some extent against this, but it would be good to think about what this means statistically (e.g. if the majority of the population in an aggregate are cheaters, if there is some frequency-dependent selection, then in most lines you would miss the cooperators).

This is an interesting question. As you said, we chose single-strain isolates so that our experiments would be reproducible, and we could correlate variation in behavior with genome sequence data. Given that our experiment is highly replicated, this approach should be quite robust to sampling error associated with selecting non-representative genotypes from these populations.

As for cheating, this is an interesting question. We generally don’t see any evidence for cheating in our system, as there are not ready public goods that are available for exploitation. Over 12 years of working on snowflake yeast, we’ve never seen a putative cheat (and indeed, cannot imagine what one would do to cheat in that system). Floc formation might be cheatable if a strain produced less Flo1 protein and yet found its way into groups of high Flo1 producers, but overall we do not see evidence for a reduction in flocculation efficiency. Quite the contrary, 19/20 of our Floc isolates evolve to flocculate more strongly, which is not consistent with cheating being common in these populations. Interestingly, flocculation was lost entirely in one population (line 20), which could have been due to pressure from a non-flocculating cheat. This is a question a new postdoc in our lab is examining, though to be clear, we do not have any data that suggests that cheats exist yet.

Reviewer #2 (Recommendations for the authors):This study has a few fundamental flaws that substantially limit what can be concluded.1) The logic and interpretation of the selection and fitness experiments are circular.1a) The logic of the experiment is circular. As demonstrated previously by the authors, settling/centrifugation-based selection overwhelmingly selects Snowflake, not Floc.However, the key points of this paper, is that using this selection regimen, they find that Snowflake is fitter than Floc under these selective conditions. Thus, the authors use a roundabout way to demonstrate that this selective regimen selects for Snowflake, not Floc, a fact already demonstrated.1b) Related to point #1a, the second issue not considered is developmental constraints. For reasons that are unclear, Floc- yeast overwhelming seems to be selected for Snowflake. Thus, the cellular arrangements that promote this morphology are part of the evolutionary fitness valley that this strain is in that is highly optimal for this selective regimen. Indeed, the authors describe this in excellent detail in the discussion (that cellular connections matter for evolving Snowflake vs. Floc). Likewise, Floc in this regimen selects for stronger Floc, which is also in a fitness valley of its own. The selection regimen pushes these further into their fitness valleys that are sub-optimal under these selective conditions. Thus, the authors are not comparing the fitness of cellular-based undifferentiated multicellular versus aggregative undifferentiated multicellularity, they instead are measuring the fitness differences of the valleys these selected strains are in (and pushed further into), which again, as in point #1a overwhelmingly selects for Snowflake, not Floc. Thus, comparing an evolved strain optimized for selective conditions versus an evolved strain sub-optimal to the selective conditions.1c) The key missing experiment is to reverse the selection. I.e. put Snowflake and Floc into a selective regimen that selects for Floc and asks which strain is fitter. However, should Floc be fitter in these conditions, it would demonstrate the null hypothesis here, which is that "you get what you select for". Without doing the experiment itself, however, interpreting the experiment as presented ends up being incomplete. This doesn't even get into more complicated selective regimens with something like predators and mouth/feeding size limits.1d) One of the key points that do seem evident is that different modes of selection for the undifferentiated multicellular act at different levels (cell vs group). However, this is largely known as different developmental mechanisms of multicellularity are known to cell vs. group level selected. This experiment does demonstrate this point de novo, though the results are not unexpected.

The first issue is that previous work has already demonstrated that selection for settling selects for Snowflake, not Floc, which makes it unsurprising that Snowflake is fitter than Floc under these same selective conditions.

We do not actually compare the fitness of snowflake to floc in this paper, and we never claim that snowflake are more fit than floc (rather, we claim that, relative to their respective snowflake and floc ancestors, snowflake evolve to significantly increase their fitness during settling selection while floc do not). Further, floc yeast do not suffer any limitation in their ability to respond to settling selection. In fact, they actually respond better to this selection than snowflake yeast do.

The point that “settling/centrifugation-based selection overwhelmingly selects Snowflake, not Floc” likely stems from Pentz et al. (2021), where we compared the floc to snowflake yeast in competition. There, we found that snowflake yeast interacted with floc in a way that gave them an advantage during settling selection by nucleating flocs around themselves. In this paper, all of our competition experiments are done in isolation. Floc and snowflake yeast never interact directly. All fitness measures are relative to that strain’s own ancestor (i.e., evolved floc vs ancestral floc, and evolved snowflake vs. ancestral snowflake). In fact, we show in this paper that floc yeast *readily* respond to settling selection. While floc and snowflake started out settling at a similar rate, over 24 weeks of evolution, floc evolve to settle much more quickly than snowflake yeast, increasing by 1200%, while snowflake only increase by 30%. See Figure 1B:

Furthermore, the fitness of evolved floc yeast greatly improves (more than snowflake!) under our experiment regime, with a ~20% average fitness increase over three days of direct competition against their own floc ancestor with daily selection for settling (third panel of Author response image 1). The central argument of our paper is that, despite the evolution of fast sedimentation and increased fitness in floc, this group-level trait did not increase their fitness during settling selection in a measurable way when competing evolved yeast against their own floc ancestor. In contrast, snowflake yeast only increased in settling speed by 30% over the same 24 weeks, but this greatly increased their fitness during settling selection when competing against their own snowflake ancestor (see first two panels of Author response image 1). This is a result of how group formation (clonal development vs. floc) changes the ability for group-level selection to impact fitness: group-level selection is far more efficient in clonal groups than aggregative flocs, due to the much higher assortment of genotypes in clonal groups. Flocs form groups that contain both evolved and ancestral genotypes, so group-level selection does little to change allele frequencies. This result does not stem from confounding issues with how we did our experiment, but is a result of the difference in assortment between clonal and aggregative multicellular organisms. This is a fundamental result with broad applicability to the evolution of multicellularity, as it helps contextualize the differences in how aggregative and clonal multicellular organisms have evolved.

**Author response image 1. sa2fig1:** 

We clarified this in the main text, by adding in overall fitness data for floc and snowflake to Figure 3 (new data in the left panel of A), which makes it clear that floc is adapting more rapidly to these conditions than snowflake yeast.

2) This experiment, due to point #1, lacks generalizability. Modeling helps with generalizability, but I think empirical data is critical and easily obtainable here.2a) Part of the way to address this would be by addressing point 1 above.

Point 1 (and thus Point 2) is not an actual issue with our paper, but rather a misunderstanding on the part of the referee.

2b) The most logical experiment would be to repeat these experiments in a related system "multicellular" system under different selective conditions and ask which morphology is fitter. Some of the authors have previously selected multicellular *Chlamydomonas*. Or transitioning the experiments to *Dictyostelium* and selecting at different stages of its life cycle. One or both of these would be highly powerful in generalizing the potential findings.

While we would love to see the development of other model systems to address these important evolutionary questions, the experiments we conducted here cannot be done in any other model system. To the best of our knowledge, it is not currently possible to engineer otherwise isogenic genotypes of *Chlamydomonas* or *Dictylostelium* to form clonal groups or aggregative groups. Even if one knew how to do this, to engineer and evolve a different model system is surely the purvey of another paper- not a condition of acceptance for this paper. It would be, at a minimum, several years of work and several hundred thousand dollars of research costs, and only if it were even possible to do. This is a ‘what about-ism’ that is not germane to our paper.

3) What happens when selection is relaxed?3a) One of the claims is that resources during growth vs. cell division are where selection is operating, but what happens when this is relaxed? Loss of the undifferentiated multicellularity at different rates when selection is relaxed would be highly informative as to which strain is fitter.

Understanding how simple multicellularity is lost under relaxed selection is an interesting question, but this would not say anything about which strain is fitter. It is a different question entirely. To understand the fitness of floc and snowflake yeast, you have to measure fitness. Which we did, in Figure 3.

3b) One of the key things would be that in many organisms, motility will play a very large role in selection. As most unicellular Eukaryotes are motile (and their multicellular relatives – e.g. Chlamy and Dicty above), thus the fitness of group vs. individual may be substantially different when motility is taken into account.

Motility can be important for some multicellular organisms, but it is not part of our well-controlled evolution experiment. This is a ‘what about-ism’ that is not germane to our paper.

3c) One potential aspect of this is the balance of epigenetic vs. genetic changes given the results of the genome re-sequencing analysis. I.e. group level trains may be selected at the genetic level, while epigenetic traits may be selected at the cellular level or vice-versa thereby complicating the conclusion of this work. The authors do indeed directly state this in lines 444-445 "While floc and snowflake yeast differ starry in evolutionary mode, our experiment was too short to examine the long-term evolutionary consequences". Which, if these facts are unknown, makes interpretation of the findings in this paper difficult to place especially given the nature of only examining one selective condition.

Given that we do not attempt to disentangle the genetics of cell vs group-level adaptation in this paper (and doing so would be very non-trivial), adding in epigenetic data would not change our conclusions. This is a ‘what about-ism’ that is not germane to our paper.

Reviewer #3 (Recommendations for the authors):L. 92: Given that you do not explain yet that you are selecting for sinking, this sentence comes a little out of the blue (there are many other possible drivers of multicellularity). And given the start of the next paragraph, maybe it makes sense to specify that size selection is performed through settling.

Good call, we edited this at line 88 to introduce the concept of settling selection earlier in this paragraph. The revised text reads:

“We evolved 20 populations of each genotype for 24 weeks with galactose as the main carbon source, ensuring robust flocculation, selecting daily for both faster growth and increased multicellular size by selecting for rapid sedimentation in liquid media^37^... While buoyancy regulation may be an important driver of multicellularity in some lineages^39^, we chose this selective regime because it is an efficient way to select on multicellular size, a fundamentally important multicellular trait^40^.”

L. 147 How is this affected by cell density – I understand that you are using the coefficient of variation to standardize over different densities, but I wonder if your pixel opacity saturates at a certain density (this is not a problem in comparing your ancestor to evolved because the differences are very clear, but it could be a problem between similar looking flocculation efficiencies). I wonder if instead, you could normalize pixel heterogeneity by cell number in some way.

Great point! There may be a degree of pixel saturation in regions with high cell densities (e.g., really dense flocs). Thus, we generated a histogram of pixel brightness from representative images of the two flocciest isolates (populations F7 and F18, see Figure 2G). The pixel opacity doesn’t seem to saturate at high pixel values representing cells in tight flocs or low pixel values representing cells in the planktonic culture, suggesting we have dynamic range to determine flocculation efficiency utilizing this approach, see panel C in Figure 2—figure supplement 2.

Additionally, cell densities are similar in each population as they are taken from stationary phase cultures (an average of 4.1x10^7^ cells/mL with a standard deviation of 1.3x10^7^ cells/mL; see first panel A in Figure 2—figure supplement 2). Furthermore, there is not a strong correlation between cell density at stationary phase and flocculation efficiency (y=6e^-10^x + 0.0649, R^2^=0.0481, *p*=0.34, panel B in Figure 2—figure supplement 2).

We added the following to the manuscript at line 153:

“We confirmed that increased flocculation was not a product of cell density in floc cultures (Supplementary Figure 5A,B).”

Figure 2. Change the ancestor reference line to have smaller dashes, or make it a different color. Dashes are the same width as the error bars and very similar to the width of the bars in the bar plot, making it visually harder to read the plot.

Great, we have done so. Thanks!

Line 534. How robust is the algorithm to bin size?

Great question! We’ve run a sensitivity analysis by analyzing the flow data with various bin sizes. The algorithm is quite robust to bin size, and mean cluster size of the biomass-scaled ancestor and evolved snowflake yeast are not significantly affected by changing bin size (see Figure 2—figure supplement 4B). We have included this sensitivity analysis in the supplement.

It is not clear to me how competitions of snowflakes were performed – cell counts on a flow-cytometer? How did you perform the counts?

Snowflake competitions were performed by directly competing a GFP-tagged ancestor with non-GFP evolved isolates. As snowflake yeast cannot be broken up into individual cells, counts of GFP-tagged and non-GFP clusters, not cells, were obtained on a flow cytometry where they can be distinguished via the FL1 fluorescence channel. We have edited the text to make this more clear, pasted below, at line 576:

“For snowflake competitions, whole cluster counts of the GFP-tagged ancestor and evolved isolate were obtained via flow cytometry using a CyFlow Space flow cytometer where GFP and non-GFP clusters can be distinguished using the FL1 fluorescence channel. For floc competitions, flocs were deflocculating using 50 mM EDTA (pH 7) prior to running on the flow cytometer, and cell counts of the GFP-tagged ancestor and evolved isolate were obtained similar to snowflake competitions.”

Line 328 – explain better why this is the case

We have motivated why the hypergeometric distribution is appropriate by explaining how our model equates settling selection to random sampling without replacement. The relevant modified text is:

“As a point of comparison we begin by considering a simple population of equal-sized groups in which each group contains exactly one mutant cell. In this scenario if there are *k* mutant cells then there are also *k* groups with a mutant cell. Since all groups are the same size, we can model the process of selection as a statistical sampling procedure in which groups are sampled randomly without replacement. For the mutant lineage to go extinct none of the *k* groups containing a mutant can be The probability of such an event (call it *pe*) is described by a hypergeometric distribution: ∏i=0fNN−k−iN−i, where the total number of groups selected is expressed as a fraction *f* of a total population of *N* groups.”

This is beside the point, but I found interesting the last sentence… how can one get at first principles from fieldwork? Fieldwork helps us constrain the parameter space to what happens in nature, but I do not know if it is possible to get first principles when many things are all varying at once.

Sorry, the first-principles here (see sentence copied below) was an adjective modifying just theory! We agree that this would be a strange adjective to apply to fieldwork, but it does play an indispensable role in a holistic understanding of multicellular evolution. To be honest, we wanted this final sentence to be inspirational and inclusive, and say that we need a variety of approaches together to understand this topic. We hope this came across!

“Developing a robust, bottom-up understanding of multicellular evolution from first-principles theory will require the integration of multiple approaches- including mathematical modeling, fieldwork, and experiments incorporating both synthetic and naturally-evolved multicellular organisms.”